# A Comparative Evaluation of Mediastinal Nodal SUVmax and Derived Ratios from ^18^F-FDG PET/CT Imaging to Predict Nodal Metastases in Non-Small Cell Lung Cancer

**DOI:** 10.3390/diagnostics13071209

**Published:** 2023-03-23

**Authors:** Maha AlRasheedi, Sai Han, Helene Thygesen, Matt Neilson, Fraser Hendry, Ahmed Alkarn, John D. Maclay, Hing Y. Leung

**Affiliations:** 1School of Cancer Sciences, University of Glasgow, Glasgow G61 1QH, UK; 2286192A@student.gla.ac.uk (M.A.);; 2West of Scotland PET Centre, Gartnavel General Hospital, NHS Greater Glasgow and Clyde, Glasgow G12 0YN, UK; 3Cancer Research UK Beatson Institute, Garscube Estate, Switchback Road, Glasgow G61 1BD, UK; 4Department of Respiratory Medicine, Glasgow Royal Infirmary, NHS Greater Glasgow and Clyde, Glasgow G4 0SF, UK

**Keywords:** non-small cell lung cancer, ^18^F-FDG PET/CT, lymph node, SUVmax, staging

## Abstract

**Simple Summary:**

Lung cancer is a major cause of premature death worldwide. The majority of lung cancers are considered non-small cell lung cancer (NSCLC). Positron emission tomography with computed tomography (PET/CT) is part of routine clinical staging, but inadequate accuracy in detecting cancer invasion in nearby lymph nodes prevents optimal surgical treatment. To improve the objectivity of PET/CT reporting, we analysed objectively obtained measurements of signals from suspicious lymph nodes inside the chest. We further obtained derived (corrected) values based on the respective measured signals of the lung tumour and blood within the core tissues in the chest and liver as well as the size of the lymph nodes. We found that the use of objectively measured signals improves the accuracy of cancer detection in the lymph nodes with potential improvement observed when using derived values (ratios). Future research may test the value of our findings in routine clinical practice.

**Abstract:**

^18^F-FDG positron emission tomography with computed tomography (PET/CT) is a standard imaging modality for the nodal staging of non-small cell lung cancer (NSCLC). To improve the accuracy of pre-operative staging, we compare the staging accuracy of mediastinal lymph node (LN) standard uptake values (SUV) with four derived SUV ratios based on the SUV values of primary tumours (TR), the mediastinal blood pool (MR), liver (LR), and nodal size (SR). In 2015–2017, 53 patients (29 women and 24 men, mean age 67.4 years, range 53–87) receiving surgical resection have pre-operative evidence of mediastinal nodal involvement (cN2). Among these, 114 mediastinal nodes are resected and available for correlative PET/CT analysis. cN2 status accuracy is low, with only 32.5% of the cN2 cases confirmed pathologically. Using receiver operating characteristic (ROC) curve analyses, a SUVmax of N2 LN performs well in predicting the presence of N2 disease (AUC, 0.822). Based on the respective selected thresholds for each ROC curve, normalisation of LN SUVmax to that for mediastinum, liver and tumour improved sensitivities of LN SUVmax from 68% to 81.1–89.2% while maintaining acceptable specificity (68–70.1%). In conclusion, normalised SUV ratios (particularly LR) improve current pre-operative staging performance in detecting mediastinal nodal involvement.

## 1. Introduction

Non-small cell lung cancer (NSCLC) is the leading cause of cancer-related death worldwide [1,2]. Timely diagnosis along with the accurate staging of the disease in patients presenting with NSCLC is crucial in deciding the optimal therapeutic approach [3,4]. The assessment of regional lymph nodes (LN) for metastases using non-invasive and invasive modalities is an integral component of this staging process [5]. In selected NSCLC patients with mediastinal lymph node disease (N2), surgical resection is recommended with curative intent [6]. However, these patients are particularly at risk of occult LN metastases and therefore increased risk of cancer recurrence [7].

Positron emission tomography/computed tomography (PET/CT) scanning combines the excellent morphological data of CT and superior sensitivity of PET imaging to assess nodal metastases. FDG (fluorodeoxyglucose) PET imaging is well established as the standard of care for staging patients with NSCLC prior to treatment with radical intent [8,9,10]. However, even with PET/CT imaging, the detection of malignant lesions remains challenging, reflecting an urgent need for improvement in terms of sensitivity and specificity [8,11]. In addition, a standardised method of reporting PET/CT scans in lung cancer staging has not been established.

The standardised uptake value (SUV) is a relative measure of radiotracers, particularly FDG uptake in routine PET imaging [12]. The performance of SUV as a determinant of nodal metastases may be affected by multiple confounding factors including serum glucose concentrations, patient body size, and the degree of local inflammation/infection as well as tumour pathologic type, underlying tumour heterogeneity, and associated cancer metabolism (e.g., rate of glycolysis). Equally, a SUV value can be affected by the signal-to-noise properties of the PET scanner; the accuracy of correction algorithms and image construction algorithms; and the duration between tracer injection and image acquisition [13,14]. 

To minimise the impact of confounding factors, previous publications have considered the normalisation of the SUV of LNs of interest to different parameters, including the size of tumour or LN as well as the SUVmax values of the primary tumour, background liver, or mediastinum. For instance, the SUV ratio of LN to primary tumour was reported to improve accuracy in predicting nodal malignancy when compared with LN SUVmax alone [13]. Here, we perform a comparative analysis of four different LN SUV normalisation methods as a means to assign risk scores to individual patients and examine their accuracies in detecting N2 nodal metastases in patients with NSCLC. The four ratios are objectively compared to investigate which normalisation approach performs best in predicting the presence or absence of mediastinal nodal metastasis.

## 2. Materials and Methods

### 2.1. Patient Cohort

In the study period of 1 January 2015 to 31 December 2017, consecutive patients with NSCLC receiving surgical resection as an initial intervention in the West of Scotland (encompassing 11 hospitals) were identified. Patient data were collected locally by clinical audit staff of each health board in accordance with the nationally recognised Quality Performance Indicator dataset and definitions. Patients included had undergone surgical resection (lobectomy, pneumonectomy, or wedge resection) along with recommended nodal dissection based on clinical staging. The pre-operative FDG PET/CTs within three months of surgery were compared with surgical nodal pathology. The clinical (pre-operative) stage was determined at multi-disciplinary team meetings based on pre-operative staging modalities, including both non-invasive imaging (e.g., CT and PET/CT) and invasive mediastinal staging (e.g., endobronchial ultrasound guided biopsies). The assessment of FDG uptake by nuclear medicine physicians and radiologists was qualitative and reported as mild, moderate, or intense for the lymph nodes of interest. The post-operative stage was based on the pathological examination of the resected tumour and lymph nodes. None of the patients received neo-adjuvant therapy prior to the PET/CT imaging. In summary, 1087 patients underwent surgical resection for their NSCLC in the study period of 2015–2017. Among these patients, 53 patients (29 women and 24 men, mean age 67.4 years, range 53–87) had pre-operative evidence of mediastinal nodal involvement (cN2).

### 2.2. ^18^F-FDG PET/CT Imaging and Image Analysis

^18^F-FDG PET/CT scans were performed according to our departmental standard operating procedure based on the established guidelines [15]. In brief, patients were fasted ≥ 4 h before intravenous injection of 400 MBq ^18^F-FDG. Blood glucose levels were measured before FDG injection to ensure levels ≤ 11 mmol/L (200 mg/dL). Patients were scanned on one of two PET/CT scanners (Discovery 690 or 710, General Electric System, Milwaukee, WI, USA) at 1 h after FDG injection. Low-dose unenhanced CT images were obtained from the skull base to mid femur level followed by PET scanning of the same levels taking 3–4 min per bed position. PET attenuation correction was based on the CT data and images were corrected for scatter and iteratively reconstructed using time-of-flight and SharpIR on a 192 × 192 matrix. PET/CT images were retrospectively analysed on GE Advantage Workstation by two PET/CT reporters (SH and FH; nuclear medicine physician and radiologist) who were blinded to nodal pathology. 

The following data from PET/CT imaging were recorded for analysis: primary tumour location, size (longest dimension), and SUVmax; mediastinal nodal locations, short axis diameter, morphology, and SUVmax; mediastinal blood pool SUVmax and liver SUVmax. SUVmax was obtained from the most FDG-avid voxel within a region of interest, signifying maximum standardised uptake value [16]. To be included in our analysis, LNs had to be anatomically defined in line with their LN station where surgery occurred, thus providing data to specify the sampled node on PET/CT. Cases were excluded if there were no visible LNs on PET/CT in the sampled LN stations. If more than one LN was visible in a single area of interest, the images pertaining to the greatest LN in that area were recorded since this was likely to be the LN included in the surgical specimen. 

### 2.3. Data Analysis

PET/CT parameters (LN SUVmax and four SUV ratios) were correlated to the nodal pathology stages. Four SUV ratios for each LN were calculated independently using the following equations:$$Tumour\;ratio\;\left(TR\right)=\frac{\mathrm{SUVmax}\;\mathrm{of}\;\mathrm{lymph}\;\mathrm{node}}{\mathrm{SUVmax}\;\mathrm{of}\;\mathrm{primary}\;\mathrm{lung}\;\mathrm{tumour}}$$
$$Liver\;ratio\;\left(LR\right)=\frac{\mathrm{SUVmax}\;\mathrm{of}\;\mathrm{lymph}\;\mathrm{node}}{\mathrm{SUVmax}\;\mathrm{of}\;\mathrm{liver}}$$
$$Mediastinum\;ratio\;\left(MR\right)=\frac{\mathrm{SUVmax}\;\mathrm{of}\;\mathrm{lymph}\;\mathrm{node}}{\mathrm{SUVmax}\;\mathrm{of}\;\mathrm{mediastinal}\;\mathrm{blood}\;\mathrm{pool}}$$
$$Lymph\;node\;size\;ratio\;\left(SR\right)=\frac{\mathrm{SUVmax}\;\mathrm{of}\;\mathrm{lymph}\;\mathrm{node}}{\;\mathrm{lymph}\;\mathrm{node}\;\mathrm{size}}$$

As an illustration, FDG PET/CT images from case #52 are presented to show the primary lesion as well as positive and negative mediastinal lymph nodes in Figure 1, which includes details of the SUVmax values along with the calculation carried out to determine the various SUVmax-derived ratios.

### 2.4. Statistical Analysis

Mann–Whitney test, chi-square test, and independent sample tests were performed using SPSS software version 28.0 (IBM Corp, Armonk, NY, USA) to compare the LN factors with related tumour data as well as the four SUV-based ratios for nodal status. *p*-values were statistically significant at the *p*  <  0.05 level. Student’s *t*-test was used to analyse normally distributed data, and Mann–Whitney U test was applied for data that were not normally distributed. Normality was assessed using a Shapiro–Wilk test (alpha = 0.05).

Receiver operating characteristic (ROC) curves were calculated using the pROC package [17]. When considering confidence intervals, it was important to note the dependency structure in our data whereby each lymph node constituted a single observation and each patient represented a cluster of observations. To retain this dependency structure, we used the block bootstrap approach described by Sherman and le Cessie [18], which accounts for unequal cluster sizes. For each predictor, we performed 100,000 bootstrap samples and then formed each confidence interval by selecting the 2.5 and 97.5 percentiles. It is notable that the percentile bootstrap appeared to be appropriate here, since the distribution of each set of bootstrap estimates was approximately symmetrical with mean relative errors of the order 0.1%. To assess the capacity of LN SUVmax, TR, LR, MR, and SR to predict nodal malignancy, ROC curves combined with the measurements of the area under the curve (AUC) were recorded. Once the ROC curves had been analysed, the cut-offs for these variables were selected that maximised Youden’s J statistic, and their respective sensitivity and specificity were determined. Comparisons between different ROC curves were carried out using the DeLong test to compare the respective AUC values and were corrected for multiple testing using Holm’s method.

## 3. Results

### 3.1. Patient Cohort and Tumour Characteristics

A total of 53 patients (29 women and 24 men, mean age 67.4 years, range 53–87) had pre-operative evidence of mediastinal nodal involvement (cN2). The location of the primary tumours was predominantly in the right upper lobe (RUL) (36%) and right lower lobe (RLL) (25%), with adenocarcinomas (55%) being the most common histopathologic type (Table 1). In total, 114 resected mediastinal nodes from the 53 cN2 patients were available for comparison between histology and retrospective PET/CT analysis with an average of 2–3 N2 nodes per patient. All 114 nodes were ‘mapped’ back to the PET/CT scans to generate the respective SUVmax values.

### 3.2. Correlative Analysis of Lymph Node Involvement

The distribution of the dissected LNs and their metastatic involvement in different stations within the mediastinum (stations 2–9) are summarised in Table 2A. In our cohort of patients with cN2 disease, only 32.5% of the mediastinal LNs were histologically confirmed to be nodal metastases post-operatively (Table 2). The physical characteristics of the mediastinal LNs are summarised in Table 2B, including mean LN short-axis lengths and primary tumour dimensions as well as the respective SUVmax values (Table 2B). The mean mediastinal LN SUVmax was 5.1 (SD 5.1).

The correlation between the PET/CT parameters of LN and LN metastases (positive LN) is summarised in Table 3, with all of the parameters analysed showing significant association with nodal metastasis. Overall, LN SUVmax-based markers were strongly associated with the presence of mediastinal (N2) metastasis. 

### 3.3. ROC Curve Comparison and Selected Threshold Values

The primary objective of this study was to evaluate SUV ratios of LNs derived from various parameters and test if they improve the accuracy of pre-operative mediastinal LN staging. The variables were examined via receiver operating characteristic (ROC) curves based on N2 nodal involvement (Figure 2). A pairwise comparison of the AUC values of the ROC curves revealed significant difference between LN SUVmax and SR (LN_SUVmax/LN_Size) as well as between MR (LN_SUVmax/Mediastinum _SUVmax) and SR (Figure 3). Primary tumour SUVmax in isolation was a poor predictor for the status of N2 LN (AUC, 0.400; 95% CI, 0.309, 0.499). Across the entire cohort of 114 N2 nodes, a SUVmax of N2 LN performed well in predicting the presence of N2 disease (AUC, 0.822; 95% CI, 0.729, 0.902), with LN_Size alone performing less well (AUC, 0.767; 95% CI, 0.677, 0.848). The normalisation of LN SUVmax by LN size (SR) (AUC, 0.658; 95% CI, 0.566, 0.759) was outperformed by the normalisation of LN_SUVmax by SUVmax values of the mediastinum, liver, and primary tumour (MR, LR and TR), respectively (Figure 2).

We further evaluated the performance of different normalisation approaches and determined their respective sensitivity and specificity based on selected cut-off values (using Youden’s J statistic to determine the highest sensitivity and specificity performance) in predicting N2 nodal involvement (Figure 4 and Table 4 and Table 5). The selected threshold of LN_SUVmax to predict nodal malignancy was 4.4, which was associated with 67.6% sensitivity and 83.1% specificity. Normalised LN SUVmax in LR, MR, and TR improved sensitivity from 68% to 81–89% while maintaining acceptable specificity (68–70%). Tumour SUVmax with a threshold at 37.9 had excellent specificity at 98.7% but poor sensitivity of only 2.7%. Interestingly, while LN size itself did not perform well, normalising LN SUVmax to LN size improved sensitivity to 81% at the expense of a drop in specificity to 47%. 

We next carried out direct comparison between the respective sensitivity and specificity based on the N2 stage in the pre-operative PET/CT reports with data derived from the determined threshold of LN_SUVmax values of the N2 nodes in Table 4. The sensitivity and specificity of PET/CT reports for cN2 disease were 66.7% and 11.6%, respectively, with comparable sensitivity but poor specificity relative to N2 staging based on LN_SUVmax values (Table 4).

## 4. Discussion

The accuracy of pre-treatment lung cancer staging is crucial to supporting evidence-based treatment decisions and predicting patient prognosis. Patients with clinical stage III disease (which includes the N2 +ve patient subgroup) account for around 10% of all patients with newly diagnosed lung cancer. Among patients with stage III disease, cN2 patients represent a particular subset that is difficult to accurately stage pre-operatively [19]. This report innovatively attempted to study this particular patient subgroup. Published data typically report on the performance of PET/CT in patient cohorts that include only radically treatable disease based on diagnostic CT scans [20,21] with patients ranging from those with T1aN0 tumours to those diagnosed with unexpected metastases on PET/CT, precluding them from radical treatment. Thus, the overall performance of PET/CT scans in staging lung cancer as documented in the literature does not directly inform the current accuracy of clinical staging for patients with N2 disease. Importantly, our cohort represents an important patient subgroup that warrants additional efforts to improve the accuracy of pre-operative staging. 

Our patient cohort focussed on a patient subgroup as part of clinical stage III resectable disease, signifying the highly relevant and timely management priority of improved staging accuracy, particularly given the importance of accurate staging to select stage II and III patients for both adjuvant and neoadjuvant immunotherapy in addition to chemotherapy so as to improve survival in both resectable and unresectable disease [22,23,24]. To our knowledge, we carried out the most comprehensive comparative analysis to date of multiple normalised LN SUVmax values (SUV ratios) on consecutive patients with pre-operative N2 NSCLC undergoing surgical resection with lymphadenectomy. With a focus on the status of N2 LN, our study was modest, comprising 53 patients and 114 N2 nodes. Additional patient cohort(s) would increase the confidence of our observations. It is worth noting that our patient cohort represented patients managed from 11 hospital sites, thus increasing the impact of our findings. However, while there were 11 hospital sites with separate multi-disciplinary team meetings to decide on the investigation and treatment of these patients, PET/CT scanning was performed at one central location, and all the patients received their operations at the same thoracic surgical centre. Thus, any inter-hospital multi-disciplinary team variability was negated in terms of the consistency of PET/CT reporting and cancer surgery. A time interval between PET/CT scan and surgery of within 3 months was set as the inclusion criteria for our study. We accept that PET/CT readings in NSCLC patients may change within this period. However, any changes that occurred during this time period would have impacted the performance of both subjective reporting within our clinical pipeline and the use of SUVmax-based PET/CT reporting. Importantly, the time interval between PET/CT scan and surgery for the vast majority of patients in our cohort was much shorter than three months (median = 30 days, mean = 41 days), while >92% of patients received their surgery within 2 months of the scan.

The focus of our study was patients with clinical (pre-operative) evidence of N2 nodal metastasis based on current staging modalities. In this study, tumour SUVmax with a threshold at 37.9 gave an excellent specificity of 98.7%, but such a threshold value is extremely high and rarely observed in clinical practice; thus, it is unlikely to be useful in routine PET imaging. SUVmax values can be affected by a number of confounding factors due to both patient specific parameters and the method employed for imaging [25], including but not restricted to serum glucose levels, body weight, and the duration between injection with radiotracer and the commencement of the imaging procedure [25,26]. 

Analysis performed in this study relied heavily on confidence in the pathologic evaluation of resected lymph nodes to provide a reference for comparative analysis. Potential bias may arise from the sub-optimal or inadequate sampling of mediastinal lymph nodes. Hence, we only included those LNs that were anatomically defined in line with the respective LN station where surgery was recorded to have occurred for analysis, thus providing data to directly relate to the sampled lymph nodes. 

Our major finding was that the use of LN SUVmax values in assessing N2 LN was more accurate than the current (qualitative) clinical practice in determining clinical evidence of N2 LN metastasis. Low specificity (11.6%) based on PET/CT reports in isolation for N2 involvement is consistent with the tendency in subjective reporting to consider equivocal signals in the N2 nodes as positive in order to not miss any nodal metastasis. Indeed, the use of invasive mediastinal staging (e.g., endobronchial ultrasound guided biopsies) is incorporated into routine clinical practice. Our data suggests that the objective reporting of PET/CT scans based on LN_SUVmax will improve the overall performance of PET imaging when incorporated into the routine clinical pipeline. It is also worth noting that despite the strong evidence of invasive mediastinal staging in determining N2 status, there remains a substantial proportion of patients not assessed in accordance with the established guidelines for the staging of N2 nodal status. The recent Dutch Lung Cancer Audit showed that guideline adherence in Dutch NSCLC patients with an indication of invasive mediastinal staging remains poor [27]. Thus, in current practice, improvement in the accuracy of staging based on imaging data remains highly impactful.

We further investigated whether normalised LN SUVmax values (SUV ratios) improved the prediction of N2 disease in comparison to unadjusted LN SUVmax or LN size. Our ROC curve comparison revealed highly comparable performance for the accuracy of N2 nodal staging based on AUC for LN SUVmax and the different derived ratios (Figure 2), with AUC values of TR, LR, MR, and SR ranging from 0.65 to 0.83. Based on the respective selected cut-off values, TR, LR, and MR improved the sensitivity of LN SUVmax alone from 68% to 81–89%. SR performs less well than the other three SUV ratios, with LR giving the highest sensitivity at 89.2% and an acceptable specificity at 68%. In this study, we analysed multiple normalisation ratios in the same patient cohort, thus extending published research on the limited number of normalised LN SUVmax ratios [28,29,30,31]. Consistent with our findings on LR, LN SUVmax to liver was shown to be associated with progression-free and overall survival in patients with NSCLC [32,33,34]. Future studies are required to test whether the adoption of LR in PET/CT imaging in clinical practice will translate into improved pre-operative accuracy in N2 nodal staging.

## 5. Conclusions

The use of LN SUVmax may improve the accuracy of identifying patients with N2 nodal disease when compared with current clinical practice. Despite the fact that respective ROC curves have similar AUC values, a comparative analysis of LN SUVmax and normalised LN SUV ratios (TR, LR, and MR) based on selected thresholds for each parameter identified potential improved performance with the use of normalised SUV ratios (particularly with LR) with a sensitivity of 89.2% in detecting N2 nodal metastasis.

## Figures and Tables

**Figure 1 diagnostics-13-01209-f001:**
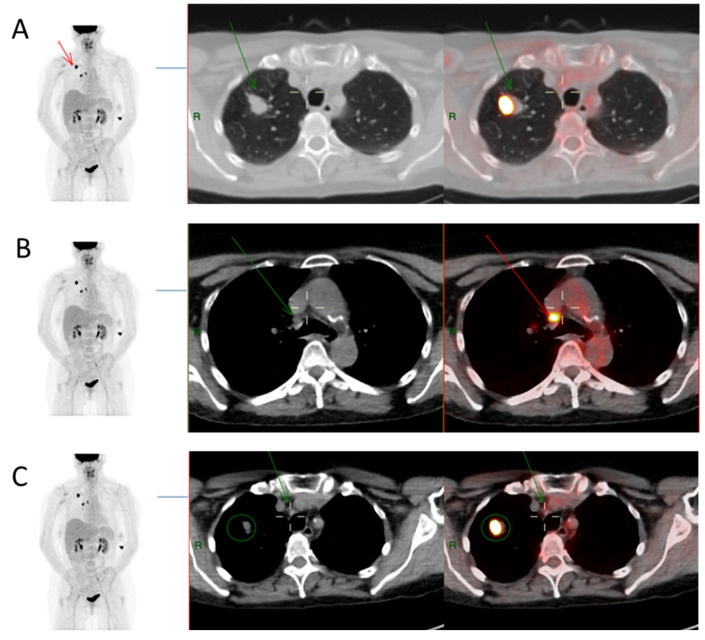
Extracted images from FDG PET/CT scan of a single patient (Case #52). (**A**) Primary tumour—FDG PET/CT images showing a right upper lobe tumour (red arrow in image on left and green arrow in Axial CT and fused PET/CT images) with measured SUVmax value of 26.1. SUVmax was also obtained from reference background structures: mediastinal blood pool (ascending aorta), SUVmax 2.9; and liver (right lobe), SUVmax 4.5. (**B**) Positive mediastinal lymph node on histology—SUVmax of mediastinal nodal station 4R (7 mm right lower paratracheal node, green arrow) was 8.0, hence SUV ratios of MR (8/2.9 = 2.8), LR (8/4.5 = 1.8), TR (8/26.1 = 0.3), and SR (8/7 = 1.1). 4R nodal pathology was confirmed to be metastatic. (**C**) the 4 mm mediastinal nodal station 2R (4 mm right upper paratracheal node, green arrow) has SUVmax value of 2.5, giving SUV ratios of MR (2.5/2.9 = 0.9), LR (2.5/4.5 = 0.6), TR (2.5/26.1 = 0.1), and SR (2.5/4 = 0.6). The 2R nodal pathology was benign. The primary tumour is illustrated by green circle in this image.

**Figure 2 diagnostics-13-01209-f002:**
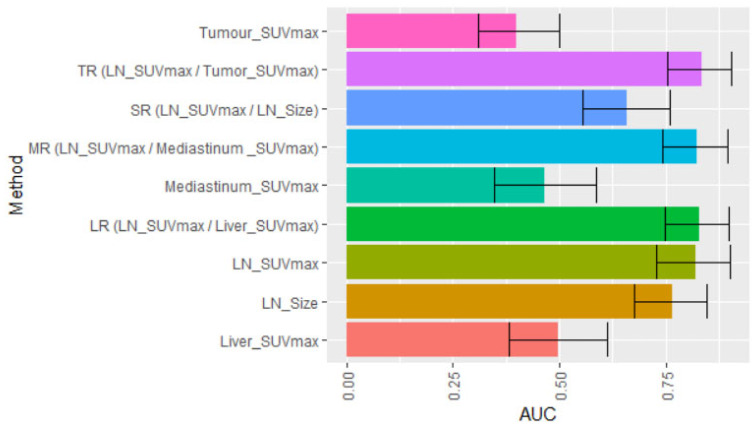
Summary of data from ROC curve analysis of individual markers for mediastinal (N2) nodal involvement and normalisation of lymph node SUVmax values (LN_SUVmax) to factors of interest. Data presented for individual bar graphs represent data on each AUC, with 95% confidence interval highlighted (ROC = receiver operating characteristic; AUC = area under curve).

**Figure 3 diagnostics-13-01209-f003:**
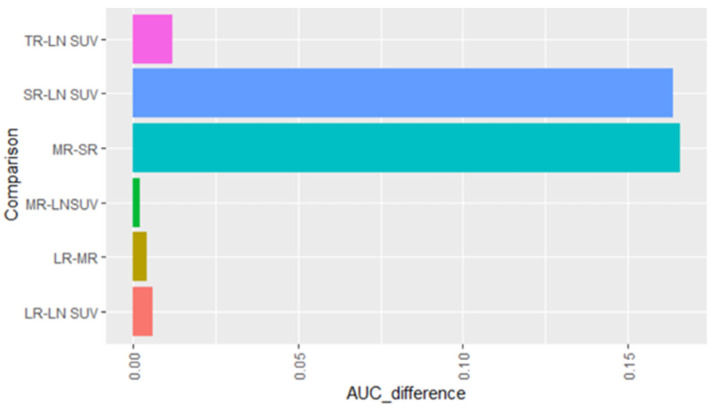
Pairwise analysis of AUC values of different ROC curves presented as bar graphs. TR = LN_SUVmax/Tumor_SUVmax; SR = LN_SUVmax/LN_Size; MR = LN_SUVmax/Mediastinum _SUVmax; LR = LN_SUVmax/Liver_SUVmax.

**Figure 4 diagnostics-13-01209-f004:**
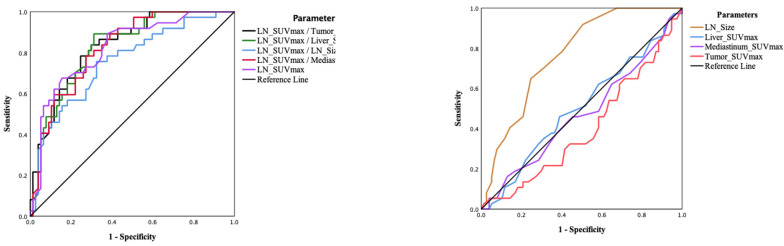
ROC curve analysis comparing the diagnostic performance of individual markers for mediastinal (N2) nodal involvement and normalisation of SUVmax of the lymph nodes (LN_SUVmax) to factors of interest. The left-side panel shows ROC curves comparing various normalisation methods (ratios) to LN SUVmax, and the right-side panel shows the ROC curves for the other parameters examined.

**Table 1 diagnostics-13-01209-t001:** Clinico-pathologic characteristics of patients (7 patients did not have tumour grade recorded while 8 tumours were of less common histology type; these are grouped as ‘others’).

Variables	% (Number of Cases in Bracket)
**Gender (%)**	
Male	45.3% (n = 24)
Female	54.7% (n = 29)
**Age (years); Mean, SD (range)**	
67.52, 8.31 (53–87)	
**Tumour location (%)**	
RUL (right upper lobe)	35.8% (n = 19)
Right hilar	7.5% (n = 4)
RML (right middle lobe)	3.8% (n = 2)
RLL (right lower lobe)	24.5% (n = 13)
LUL (left upper lobe)	24.5% (n = 13)
LLL (left lower lobe)	3.8% (n = 2)
**Types of Surgical Resection (%)**	
Lobectomy	77.4% (n = 41)
Lobectomy + wedge resection	13.2% (n = 7)
Pneumonectomy	7.5% (n = 4)
Wedge resection	1.9% (n = 1)
**Tumour histology (%)**	
Adenocarcinoma	54.7% (n = 29)
Squamous cell carcinoma	30.2% (n = 16)
Others	15.1% (n = 8)
**Tumour grade (%)**	
Moderately differentiated	60.4% (n = 32)
Poorly differentiated	26.4% (n = 14)
Unknown	13.2% (n = 7)

**Table 2 diagnostics-13-01209-t002:** Lymph node characteristics and associated correlations.

(A) Nodal stations and pathologic status	
**Mediastinal lymph node stations; % (number, n)**	
**Upper paratracheal (station 2)**	4.4% (n = 5)
Pre-vascular and retro-tracheal (station 3)	0.9% (n = 1)
Lower para-tracheal (station 4)	28.9% (n = 33)
Sub-aortic (station 5)	9.6% (n= 11)
Para-aortic (station 6)	4.4% (n= 5)
Sub-carinal (station 7)	35.1% (n= 40)
Para-esophageal (station 8)	9.6% (n= 11)
Pulmonary ligament (station 9)	7.0% (n= 8)
**Mediastinal lymph node histology; % (number, n)**	
Negative mediastinal	67.5% (n = 77)
Positive mediastinal	32.5% (n = 37)
**(B) Summary of parameters associated with pathologically involved lymph nodes**	
**Parameters**	**Mean, SD**
**Mediastinal lymph nodes**	
LN size (short axis dimension, mm)	8.2, 5.3
LN SUVmax	5.1, 5.1
**Other parameters**	
Tumour SUVmax	16.7, 8.21
Liver SUVmax	4.0, 0.83
Mediastinum SUVmax	2.6, 0.56
Tumour size (longest dimension, mm)	37.10, 22.71
Tumour pathological size (largest dimension, mm)	42.61, 26.21

**Table 3 diagnostics-13-01209-t003:** Correlation of nodal parameters with metastatic N2 status (positive or negative signifies the presence or absence of metastatic disease in the lymph nodes based on histopathologic assessment, respectively).

Characteristics (Mean ± SD)	Positive	Negative	*p*-Value
LN size (short axis dimension, cm)	0.98 ± 0.18	0.76 ± 0.25	<0.001
SUVmax of LN	8.22 ± 6.45	3.62 ± 3.36	<0.001
SR (LN SUV/LN size)	7.91 ± 5.36	4.87 ± 4.25	<0.001
LR (LN SUVmax/liver SUVmax)	2.15 ± 1.72	0.98 ± 1.10	<0.001
MR (LN SUVmax/mediastinum SUVmax)	3.36 ± 2.71	1.49 ± 1.70	<0.001
TR (LN SUVmax/tumour SUVmax)	0.56 ± 0.36	0.23 ± 0.20	<0.001

**Table 4 diagnostics-13-01209-t004:** Performance at selected threshold for each parameter.

Parameters	Threshold	Sensitivity (%)	Specificity (%)
LN_SUVmax	4.4	67.6	83.1
LN_Size (mm)	5.5	91.9	49.4
Tumor_SUVmax	37.9	2.7	98.7
Mediastinum_SUVmax	3.1	16.2	87.0
Liver_SUVmax	4.0	46.0	59.7
SR (LN_SUVmax/LN_Size)	0.4	81.1	46.8
LR (LN_SUVmax/Liver_SUVmax)	0.8	89.2	67.5
MR (LN_SUVmax/Mediastinum_SUVmax)	1.3	81.1	68.8
TR (LN_SUVmax/Tumor_SUVmax)	0.2	83.8	70.1

**Table 5 diagnostics-13-01209-t005:** Pairwise comparison of sensitivity and specificity in predicting N2 nodal involvement based on selected thresholds for each parameter (see Table 4) using the McNemar test. TR = LN_SUVmax/Tumor_SUVmax; LR = LN_SUVmax/Liver_SUVmax; SR = LN_SUVmax/LN_Size; MR = LN_SUVmax/Mediastinum_SUVmax.

	*p*-Values (with Adjusted *p*-Values) for Sensitivity and Specificity	
Comparison	Sensitivity	Specificity
TR-LR	0.083 (0.332)	0.157 (1.000)
TR-SR	<0.001 (0.005)	0.317 (1.000)
TR-MR	0.157 (0.471)	0.157 (1.000)
LR-SR	<0.001 (0.005)	0.317 (1.000)
LR-MR	0.317 (0.634)	0.157 (1.000)
SR-MR	<0.001 (0.005)	0.317 (1.000)
LR-LN SUV	<0.001 (0.005)	0.157 (1.000)
TR-LN SUV	<0.001 (0.005)	0.157 (1.000)
SR-LN SUV	0.317 (0.634)	0.157 (1.000)
MR-LNSUV	<0.001 (0.005)	0.157 (1.000)

## Data Availability

Not applicable.

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
