# Peer review of "A Comparative Evaluation of Mediastinal Nodal SUVmax and Derived Ratios from 18F-FDG PET/CT Imaging to Predict Nodal Metastases in Non-Small Cell Lung Cancer"

_diagnostics, 2023, doi:10.3390/diagnostics13071209_

Round 1
Reviewer 1 Report
The present study: “ comparative evaluation of mediastinal nodal SUVmax and derived ratios from 18F-FDG PET/CT imaging to predict nodal metastases in non-small-cell lung cancer (NSCLC)”, aimed to improve the accuracy of preoperative staging, for which diagnostic values were compared, including four SUV indices based on the SUV values of the primary tumor (TR), mediastinal (MR) and hepatic blood pool ( LR) and the size of the nodes (SR). This study was able to demonstrate that the use of LN SUVmax could improve the accuracy of identifying patients with N2 node disease, and a potential performance improvement was identified with the use of normalized SUV indices. The study is novel and the proposal is interesting to improve the diagnosis of cancer progression before surgery. For these reasons, I believe that the article should be approved in its current form.
As for some improvements, I recommend that the acronyms be described below the tables.
Author Response
Comments and Suggestions for Authors
The present study: “ comparative evaluation of mediastinal nodal SUVmax and derived ratios from 18F-FDG PET/CT imaging to predict nodal metastases in non-small-cell lung cancer (NSCLC)”, aimed to improve the accuracy of preoperative staging, for which diagnostic values were compared, including four SUV indices based on the SUV values of the primary tumor (TR), mediastinal (MR) and hepatic blood pool ( LR) and the size of the nodes (SR). This study was able to demonstrate that the use of LN SUVmax could improve the accuracy of identifying patients with N2 node disease, and a potential performance improvement was identified with the use of normalized SUV indices. The study is novel and the proposal is interesting to improve the diagnosis of cancer progression before surgery. For these reasons, I believe that the article should be approved in its current form.
As for some improvements, I recommend that the acronyms be described below the tables.
Response – We thank the Reviewer for this suggestion and have updated the legend for tables accordingly.
Reviewer 2 Report
The authors did a thorough study on the evaluation of the proposed improvement in PET/CT imaging strategy for patients with non-small cell lung cancer (NSCLC). The authors proved that normalization of lymph nodes standard uptake values (LN SUVmax) for each receiver operating characteristic (ROC) curves could significantly increase the sensitivities of current detection methods in detecting mediastinal nodal involvement. Overall, the manuscript did a comprehensive investigation and was able to propose a better analysis method regarding NSCLC detection. I suggest minor revisions. Here are some comments for the authors.
1. To better help readers who are not familiar with PET/CT analysis, the authors may add one example of original data from PET/CT as well as the ROC curve which SUVmax calculated from to walk through readers with the analysis steps.
2. It would be more straightforward to provide column graphs for the comparison tables, especially the one comparing the improved analysis method vs the traditional method.
3. Could the authors provide more information regarding the four SUV ratios? Why these four ratios are important in correction in ROC curves?
Author Response
Comments and Suggestions for Authors
The authors did a thorough study on the evaluation of the proposed improvement in PET/CT imaging strategy for patients with non-small cell lung cancer (NSCLC). The authors proved that normalization of lymph nodes standard uptake values (LN SUVmax) for each receiver operating characteristic (ROC) curves could significantly increase the sensitivities of current detection methods in detecting mediastinal nodal involvement. Overall, the manuscript did a comprehensive investigation and was able to propose a better analysis method regarding NSCLC detection. I suggest minor revisions. Here are some comments for the authors.
- To better help readers who are not familiar with PET/CT analysis, the authors may add one example of original data from PET/CT as well as the ROC curve which SUVmax calculated from to walk through readers with the analysis steps.
Response – We thank Reviewer 2 for this nice suggestion. We have updated the manuscript and include images to illustrate how the ratio were calculated from respective SUVmax values.
- It would be more straightforward to provide column graphs for the comparison tables, especially the one comparing the improved analysis method vs the traditional method.
Response – It is a nice suggestion which we have adopted accordingly.
- Could the authors provide more information regarding the four SUV ratios? Why these four ratios are important in correction in ROC curves?
Response – We thank Reviewer 2 for this comment. The four ratios were compared to investigate which approach of normalisation would perform best in predicting the presence or absence of mediastinal metastasis. This point is clarified in the introduction section.
Reviewer 3 Report
I congratulate the authors, the work is an excellent example of good scientific work. Elegant, well structured and scientifically valid and interesting. I have only a few minor considerations.
Minor questions:
1. In lines 65 to 69 the authors point out that the standardized uptake value (SUV) can be influenced by variables such as blood glucose concentration, body weight, inflammation state, etc. In my opinion, the authors, for the sake of completeness, should indicate in the materials and methods section the average weight of the subjects studied and the average glycaemia of the subjects studied and, if possible, also the C-reactive protein and erythrocyte sedimentation rate values. Perhaps adding these data to table 1. If these data are available and it is possible to find them.
2. Again for completeness and to enrich the work, I would suggest to the authors to indicate, in the materials and methods section, if some patients were subjected to immunotherapy (PD-1/PD-L1 axis inhibitors) and to indicate, in the discussion section, if the any immunotherapy may in some way influence the results they obtain, in particular with reference to the LN SUVmax and SUV ratios. A few lines of comment are enough.
3. Did the authors test a lung cancer marker such as CYFRA 21-1 in their patients? if so, it would be interesting to indicate the average values ​​(in the table) and indicate whether there is any correlation between this marker and the data obtained from them.
Author Response
Minor questions:
- In lines 65 to 69 the authors point out that the standardized uptake value (SUV) can be influenced by variables such as blood glucose concentration, body weight, inflammation state, etc. In my opinion, the authors, for the sake of completeness, should indicate in the materials and methods section the average weight of the subjects studied and the average glycaemia of the subjects studied and, if possible, also the C-reactive protein and erythrocyte sedimentation rate values. Perhaps adding these data to table 1. If these data are available and it is possible to find them.
Response – We thank Reviewer 3 for this interesting point. Unfortunately, we do not have the full data to support the necessary analysis.
- Again for completeness and to enrich the work, I would suggest to the authors to indicate, in the materials and methods section, if some patients were subjected to immunotherapy (PD-1/PD-L1 axis inhibitors) and to indicate, in the discussion section, if the any immunotherapy may in some way influence the results they obtain, in particular with reference to the LN SUVmax and SUV ratios. A few lines of comment are enough.
Response – We can confirm no patients received neo-adjuvant treatment, prior to surgery.
- Did the authors test a lung cancer marker such as CYFRA 21-1 in their patients? if so, it would be interesting to indicate the average values ​​(in the table) and indicate whether there is any correlation between this marker and the data obtained from them.
Response – We have not performed any biomarker analysis on this cohort. We agree that this would be very interesting area of future research.